# The Impact of the Individual and Combined Application of Phosphorus and Sulfur Fertilizers on Potato Tuber Flavor

**DOI:** 10.3390/foods12203764

**Published:** 2023-10-13

**Authors:** Kaifeng Li, Maoxing Li, Jinhua Zhou, Huachun Guo

**Affiliations:** 1College of Agronomy and Biotechnology, Yunnan Agricultural University, Kunming 650201, China; dtllx04@163.com (K.L.); limaoxing2011@163.com (M.L.); ynnydxzjh@126.com (J.Z.); 2Yunnan Engineering Research Center of Tuber and Root Crop Bio-Breeding and Healthy Seed Propagation, Yunnan Agricultural University, Kunming 650201, China; 3Tuber and Root Crop Institute, Yunnan Agricultural University, Kunming 650201, China

**Keywords:** phosphorus and sulfur fertilizers, volatile flavor compounds, fatty acid, free amino acid

## Abstract

Sulfur and phosphorus are important plant nutrients required for potato growth, influencing the synthesis of primary metabolites that serve as the material foundation of potato flavor quality. However, little is known about the effects of sulfur and phosphorus application on potato tuber flavor. This study experimentally compared the effects of the individual and combined application of phosphorus and sulfur fertilizers on the flavor of potato tubers. The research examined the sensory characteristics of flavor under various fertilization methods, investigated changes in the types and contents of volatile flavor compounds, and conducted analyses on flavor precursor compounds. The experimental results showed that the application of phosphorus and sulfur fertilizers, either individually or in combination significantly increased the content of linoleic acid and linolenic acid. After the combined application of phosphorus and sulfur fertilizers, the starch and the reducing sugar content also significantly increased. (E,E)-2,4-Nonadienal and Decanal are closely correlated with fatty acid content. Dimethyl sulfide and Trimethyl sulfide contents are significantly related to methionine content. This also significantly enhances the fatty taste characteristics of the tubers but weakens the potato flavor characteristics. Hence, the application of phosphorus and sulfur can affect the accumulation of primary metabolic products in tubers, thereby affecting flavor quality. Compared with the individual application of phosphorus or sulfur fertilizers, when phosphorus fertilizer is applied at 180 kg·ha^−1^ and sulfur fertilizer at 90 kg·ha^−1^ in combination, it can further enrich the roasted flavor characteristics of potatoes and maximize the enhancement of potato flavor quality. This provides valuable theoretical support for achieving high-quality agricultural development.

## 1. Introduction

The drastic deterioration of food security has been brought to the forefront of societal attention. According to a report by the United Nations Food and Agriculture Organization, the global incidence of food insecurity increased from 25.3% in 2019 to 29.6% in 2022 [1]. The fertilizer production sector, which is heavily reliant on energy, has been identified as a significant contributor to this burgeoning crisis [2]. Consequently, it is imperative to adopt a rationalized fertilization management strategy, characterized by the minimization of fertilizer wastage and the optimization of agricultural productivity, as a viable approach to bolster food security [3].

Central to this discourse is the acknowledgment of the critical role played by nutrients such as nitrogen, phosphorus, potassium, and sulfur in facilitating optimal plant growth and development. Historically, the primary objective of in-field fertilizer management was to enhance crop yield incrementally. However, this practice has come under scrutiny, with emerging viewpoints suggesting a potential compromise in crop quality concomitant with yield augmentation. A notable example is the observed inverse relationship between oil and protein content in rapeseed; an increase in the former seems to inhibit the synthesis of the latter, a phenomenon documented in previous studies [4]. Therefore, an intricate examination of the repercussions of fertilizer utilization on crop quality is necessary, fostering a balanced approach to yield and quality optimization.

Potatoes (*Solanum tuberosum* spp. *andigena*) stand as one of the primary staple crops globally. In the past, much of the research has been centrally fixated on scrutinizing the effects of fertilization on yield [5,6,7,8]. Regarding quality attributes, it has been elucidated that the application of potassium [9] and boron fertilizer [10] can markedly elevate the starch content in tubers, while simultaneously fostering the production of organic acids. Furthermore, nitrogen fertilization has been linked to modifications in tuber nitrate levels [11] and acrylamide content [12].

Consumer preference is heavily influenced by the flavor profile of the product, a multifaceted sensory characteristic constituted by aroma, taste, and other integral factors [13]. Consequently, achieving a flavor profile aligning with consumer expectations is deemed a quintessential attribute for commercial success in the potato industry [14]. The flavor profile, an amalgamation of aromatic compounds and gustatory characteristics, plays a pivotal role in delineating the quality of potatoes [15]. A plethora of volatile compounds, including aldehydes, alcohols, ketones, esters, acids, furans, pyrazines, and ethers, have been identified as direct contributors to the aromatic attributes, showcasing significant variation in composition depending on the culinary preparation method implemented [16].

For instance, van Loon et al. [17] identified 122 compounds in fried potatoes, including 2- and 3-Methylbutanal, whereas the number soared to 182 and over 400 in boiled and roasted potatoes respectively, encapsulating compounds like (E)-2-Nonenal, Methional, and 2,3-Diethyl-5-methylpyrazine [18]. The flavor composition is highly dependent on the cooking methods applied, primarily owing to the variations in thermal effects and moisture alterations in different culinary environments [19,20].

Volatile compounds contribute differently to the overall flavor quality of potatoes under varying cooking conditions. Methional is an essential volatile compound that presents a characteristic potato flavor. Efforts to enhance its flavor by increasing its precursor amino acid content through genetic engineering have been attempted [21]. Bough and colleagues’ study suggested that the potato-like flavor it presents contributes to the formation of a positive flavor [22]. In contrast, Zhao et al. [23] believed that the pungent odor produced by methional after potato steaming was due to its overcooking. Due to the flavor-masking effect, in some cases, its presence in fries might be imperceptible [24]. (E,E)-2,4-Decadienl, primarily produced by the cleavage of hydrogen peroxide, is typically present at lower levels than alkanals. Its roasted taste contributes to the flavor characteristics of cooked purple potatoes [25]. In fries, this compound imparts a frying flavor, and its concentration can influence the overall sensory evaluation [26]. In boiled potatoes, the content of Hexanal is believed to be related to the overall aroma [27]. Zhao et al. [28] found that during rapid high-temperature cooking, hexenal contributed to the flavor of stir-fried shredded potatoes. In fries deep-fried in highly oxidized oil, its further accumulation with compounds like E-2-Undecenal enhances the rancid flavor characteristics [26]. Apart from lipid oxidation, Maillard reaction products involving sugars and amino acids usually correlate with positive flavors, where pyrazine compounds like 2-Ethyl-3,6-dimethylpyrazine and 3-Ethyl-2,5-dimethylpyrazine contribute to the nutty, baked, and buttery flavor characteristics in fries and roasted potatoes [29]. However, Duckham found that when the pyrazine-to-aldehyde ratio in the matrix was imbalanced, these compounds contributed musty and earthy flavors [30]. Sharma et al. [13] believed that consumer groups with different orientations have preferences for potatoes’ sensory attributes that determine their consumption tendencies. Thus, when exploring the effects of different factors on the synthesis of key compounds, it is crucial to simultaneously consider their sensory characteristics.

It is noteworthy that the content of amino acids, sugars, and fatty acids, the primary precursors to these volatile compounds, can be significantly influenced by variations in cultivation management, hence suggesting a potential impact on the flavor profile.

Research spearheaded by Jansky et al. [31] uncovered an adverse effect on flavor resulting from nitrogen fertilization, primarily attributed to the significant correlation between nitrate accumulation in tubers and undesirable flavor notes [32]. Additionally, sulfur fertilization can facilitate the synthesis of sulfur-containing amino acids, playing a crucial role in nitrogen assimilation, whereas insufficient phosphorus fertilization can elevate reducing sugar levels, concurrently affecting free amino acid content [33]. However, limited literature exists on its subsequent influence on the formation of flavor compounds. Compared to other crops, potatoes demonstrate a higher demand for phosphorus fertilization [34], thus necessitating comprehensive studies to understand the interplay between phosphorus fertilization, fatty acid content, and flavor profile [35].

In light of the above, the aim of this study is to undertake a meticulous evaluation of the flavor profiles of potato tubers subjected to the individual and combined application of phosphorus and sulfur fertilizers. The overarching objectives of this investigation encompass (1) the analysis of primary metabolite alterations under different fertilization treatments, (2) the identification of volatile compounds influenced by fertilization effects, and (3) the assessment of the impact of fertilizer application on the flavor quality of potatoes. By unraveling the intricacies of the interplay between different fertilization strategies and the formation of flavor compounds, this study seeks to deepen our understanding of the nexus between food flavor chemistry and plant nutrition, thereby offering a valuable theoretical foundation for fostering a trajectory of sustainable and high-quality agricultural advancement.

## 2. Materials and Methods

### 2.1. Material Cultivation and Sample Collection

The experiment was conducted from March 2021 to August 2023 in Kunming, Yunnan Province, China (25°22′ N, 102°44′ E). The trial set up three plots (replicates), each containing two varieties: Qing Shu 9 (QS9) and Dian Shu 23 (DS23) (*Solanum tuberosum* spp. *andigena*). The experimental treatments are set as shown in Table 1, consisting of eight levels: QS9_CK, QS9_P, QS9_S, QS9_PS, D23_CK, D23_P, D23_S, D23_PS. In each treatment, there were three replicates, each planted with nine pots.

The experiment utilized triple superphosphate (45% P_2_O_5_) as the source of phosphorus and ammonium sulfate (22% S) as the source of sulfur. All treatments were supplemented with commercial urea (46% nitrogen) at 216 kg·ha^−1^ and potassium chloride (60% K_2_O) at 30 kg·ha^−1^ as sources of nitrogen and potassium, respectively. All treatments used tuber propagation for seedlings, which were later transplanted into the experimental plots. All varieties were manually harvested in August of the same year, at which point the canopy had completely died off. The soil nutrient background of the three planting seasons is shown in Table 2. After tuber harvest, three plants were randomly selected from each replicate, mixed, and used as one replicate; each treatment had three such replicates. The tubers were quickly peeled, cut, and flash-frozen in liquid nitrogen, then stored at −80 °C. For sensory evaluation, fresh tubers were stored at 4 °C and the experiment was completed within one week.

### 2.2. Sensory Evaluation of Materials

The harvested potato samples were labeled with random two-letter identifiers (such as “AC” or “BA”). Once labeled, the samples were submitted to an evaluation team for sensory assessment. These potato samples were cleaned, air-dried, and cut into small pieces (9 cm × 5 cm × 3 cm). The air fryer (EDC-ZG5, EdenPURE, North Canton, OH, USA) was preheated to 100 °C for 10 min before placing 200 g of foil-wrapped potato pieces and heating at a constant temperature of 150 °C for 30 min, with three repeats. Afterward, half of the mixed sample from each replicate was taken for sensory evaluation.

The evaluations took place in an isolated, well-ventilated laboratory, with samples placed in 25 mL headspace vials and maintained at 40 ± 1 °C using a water bath. Evaluation criteria mainly centered around aroma, encompassing eight attributes: Fatty, Potato-like, Cardboard-like, Grass, Roasted, Flavor intensity, Off-flavor, and Overall quality. Each attribute was rated on a scale from 1 to 7, with judgments ranging from “completely imperceptible” to “maximum intensity”. The sample evaluation order was completely randomized and double-blind. After completion, scores collected on evaluation sheets were used to calculate sensory characteristics.

This study was conducted with the approval of the Ethics Committee of Yunnan Agricultural University, August 2021.

### 2.3. Volatile Flavor Compound Detection

The remaining portion of the samples mentioned in Section 2.2 was used for volatile component analysis. The remaining sample had a 1–1.5 mm thick surface section cut off and thoroughly mixed. From the pre-treated materials, 3 g was taken and added to a 25 mL headspace vial with 1 µL of 2-methyl-3-heptanone (concentration: 0.163 µg/µL) (Sigma-Aldrich Pty Ltd., Darmstadt, Germany) as an internal standard. The vial was sealed with a polytetrafluoroethylene septum and equilibrated at 50 °C for 30 min in a water bath. Extraction was performed using a 75 µm CAR/PDMS (Supelco Inc., Bellefonte, PA, USA) fiber for 40 min, repeated three times for each combination. Post extraction, the fiber was desorbed in the injection port of a GC-MS (Agilent 7890A-5977C, Agilent Technologies, Santa Clara, CA, USA) instrument at 250 °C for 5 min to conduct GC-MS analysis. Specific analysis conditions were as follows: the chromatographic column was an HP-5ms (30 m × 0.25 mm × 0.25 µm) capillary column; helium was used as the carrier gas with a splitless injection and a flow rate of 1.2 mL·min^−1^. The temperature program was as follows: initial temperature 40 °C, held for 3 min, then raised at 5 °C·min^−1^ to 200 °C, followed by an increase at 10 °C·min^−1^ to 230 °C, and held for 5 min. Mass spectrometry conditions were as follows: Electron Ionization (EI) source, electron energy 70 eV, transfer line temperature 280 °C, ion source temperature 230 °C, quadrupole temperature 150 °C, a full scan mode from mass scan range (*m*/*z*) 33–550, solvent delay 1 min. Quantification and data analysis followed the method described by Bough [36].

### 2.4. Fatty Acid, Free Amino Acid, Total Reducing Sugar and Starch Detection

Fatty acid detection followed the method described by Leonova et al. [37], with further adjustments. Lyophilized potato samples (72 h at −42 °C) were ground to a uniform fine powder, with 20–30 mg placed in a 2 mL centrifuge tube. To this, 1.2 mL of chloroform–methanol solution (2:1 *v*/*v*) and 0.4 mL of 0.1 M KCl (Sigma-Aldrich Pty Ltd., Darmstadt, Germany) solution were added, followed by vigorous shaking for 3 min to allow complete reaction. After centrifugation at 10,000 rpm for 5 min, the homogenate separated into three layers. The bottom organic phase obtained through organic separation and extraction was transferred to a new tube and evaporated to dryness. The residue was dissolved in 100 µL Hexane and transferred to a gas chromatograph (GC,7890A, Agilent Technologies, Santa Clara, CA, USA) equipped with a hydrogen-flame ionization detector and a DB-Fast FAME column (30 m × 0.25 mm × 0.25 µm) for further analysis. An injection volume of 1 μL was used with a split ratio set at 80:1. Both the injection and detector were heated to 260 °C. The oven’s temperature began at 80 °C for half a minute, then climbed to 165 °C in over 1 min with an increase rate of 40 °C·min^−1^. Subsequently, the temperature escalated to 230 °C at 5 °C·min^−1^ and held steady for 4 min. Fatty acids were identified based on their retention time.

For the detection of free amino acids, 1 g of fresh potato samples was homogenized with 5 mL of 5% trichloroacetic acid. The mixture was centrifuged at 10,000 rpm for 10 min. After obtaining the supernatant, the volume was adjusted to 5 mL with 0.1 M HCL (Sigma-Aldrich). And then the samples were diluted 1 + 4 with 0.16 M lithium citrate buffer (pH 2.2, Biochrom, Cambridge, UK) and thoroughly mixed. The samples, once diluted, were passed through a 0.45 mm HPLC membrane filter before being analyzed using cation-exchange chromatography. This was subsequently followed by post-column derivatization with ninhydrin (Biochrom 30+, Biochrom, Cambridge, UK), following the protocol detailed by the manufacturer using the physiological system (Biochrom). An injection volume of 50 µL was employed, and quantification was determined by comparing it to both an external and internal standard.

The total reducing sugars were assessed using the 3,5-dinitrosalicylic acid (DNS) method as outlined by Miller [38], and the starch content was determined through the coloration technique detailed in Xu [39].

### 2.5. Statistical Analysis

For the sensory evaluation data under different fertilizer treatments, the Wilcox test was performed using the statistical package based on R (version 3.1). The content of the flavor compounds and the contents of amino acids, fatty acids, reducing sugars, and starch were tested using a *t*-test, and the results were corrected for FDR. Amino acids, fatty acids, reducing sugars, and volatile compounds were first normalized and then subjected to correlation analysis using the “psych” package. The “ropls” package was used to complete the OPLS-DA analysis of the volatile compounds and screen the key volatile flavor compounds based on the conditions of FC > 1.5, VIP > 1 and *p*-value < 0.05. And Cytoscape 3.1 to was used to draw the correlation network diagram. A principal component analysis of all data for the three planting seasons was completed using “factoextra”.

## 3. Results and Discussion

### 3.1. Changes in Tuber Sensory Attributes Due to Fertilizer Application

Figure 1 illustrates the distribution of sensory evaluation scores under four different treatments across three growing seasons for DS23 and QS9. For the characteristic labeled as “Potato-like”, overall, the scores for QS9 are slightly higher than those for DS23. Across the three growing seasons, the CK scores were higher than the other three fertilizer applications. The P treatment consistently exhibited significantly lower scores than CK, marking the lowest among the four treatments, except in the first planting season. The PS scored second lowest, significantly lower than CK in all three seasons. Comparisons between PS with P or S did not reach a significant level, indicating that the applications of the phosphorus and sulfur fertilizers both attenuated the potato flavor characteristic of the tubers, with the phosphorus-only application reducing it most prominently.

Regarding the “Roasted” attribute, DS23 and QS9 demonstrated a degree of consistency in performance. In all three planting seasons, the S and PS treatments were higher than CK, but only the PS treatment reached a significant level. It is noteworthy that the PS treatment consistently scored significantly higher than the P and S applications, highlighting the enhancing effect of the sulfur application on the roasted flavor characteristic of potatoes. The “Fatty” attribute did not have a strong representation in air-fried mature products; thus, overall, the perceived intensity for this attribute was weaker than the sensory perceptions of potato and roasted flavors. The changes under the “Fatty” attribute mirrored those under “Roasted”, indicating that fertilizer application could enhance the intensity of the “Fatty” attribute, with the most significant effect under the PS treatment. Moreover, we investigated two flavor characteristics, “Grass” and “Cardboard-like”, which might be associated with off-flavors. Notably, the “Grass” characteristic was significantly intensified only with sulfur-only application, and similarly, the “Cardboard-like” characteristic increased only when neither phosphorus nor sulfur was applied. The “Overall quality” attribute represents a broader subjective preference, thus exhibiting substantial variance in data distribution. This parameter reflects the acceptance level of the entire sensory profile by the panel of evaluators.

The roasted flavor has a positive correlation with the overall sensory evaluation [31]. Studies on fries and chips have also indicated that lipid aromas can enhance the overall flavor intensity of products [26] Thus, compared to CK, the application of fertilizer improved the tubers’ acceptance level, with PS emerging as the most favorable treatment. Comparisons among different fertilizer treatments further revealed that supplementing with one type of fertilizer, based on already applied phosphorus (P) or sulfur (S), could also enhance the overall flavor profile of the produce. In chicory studies, it was found that the application of nitrogen fertilizers could improve changes leading to a roasted flavor [40]. The use of sulfur fertilizers can elevate the plant’s nitrogen utilization efficiency, thereby promoting the synthesis of nitrogenous compounds [41]. This indirect effect might be the reason for the increase in roasted flavor after sulfur fertilizer application.

Generally, the S treatment significantly amplified roasted, fatty, grassy notes, and overall quality, while diminishing the distinct potato flavor and cardboard notes. The P treatment notably increased the roasted, fatty, and grassy flavors, overall quality, and overall flavor intensity, simultaneously weakening the potato and cardboard notes. In the PS treatment, the changes followed the same trend as the previous two treatments but further magnified the sensory characteristics observed with individual applications of phosphorus or sulfur.

### 3.2. Comparison of Volatile Compounds among Different Fertilizer Treatments

Figure 2 presents the heatmap of inter-group differences in volatile compounds under four levels of fertilizer applications, wherein a total of 83 kinds of volatile compounds were detected and quantified in this experiment. The top three categories in terms of quantity are pyrazines with 36 varieties, followed by aldehydes with 16, and alcohols comprising 10 types.

Figure 2a illustrates the comparison of volatile compounds between P and CK. Throughout the three growing seasons, over 50% of the volatile compounds in the D23 variety were upregulated, approximately 10% higher than in the QS9 variety. This potentially reflects material disparities in nutrient absorption and utilization. In the category FO based on the literature, most aldehyde compounds increased following the sole application of phosphorus fertilizer. Notably, compounds such as (E,E)-2,6-Nonadienal, 2-Methyl-furan, (E,E)-2,4-Heptadienal, and Nonanal exhibited VIP values exceeding one. Within the category MR, a portion of volatile compounds in the D23 variety elevated under this treatment, albeit displaying an overall decreasing trend. Pyrazine compounds like Benzeneacetaldehyde, 2,6-Dimethyl-pyrazine, and 2-Ethyl-3,5-dimethyl-pyrazine significantly decreased across multiple growing seasons. In the category SP, three compounds including Dimethyl trisulfide showed reduced levels, reaching a significant extent. In comparison to CK, the P treatment witnessed an increasing trend in the category FO, while both the MR and SP demonstrated a general decline.

Figure 2b delineates the comparison of volatile compounds between S and CK. Across all three growing seasons, only a quarter of the detected volatile compounds in the D23 variety were upregulated, whereas a relatively larger proportion was upregulated in the QS9 variety. In the category FO based on the literature, a portion of aldehyde compounds surged following the singular sulfur fertilizer application, with compounds like Pentanal, 2,3-Octanedione, and E-2-Nonenal exhibiting VIP values greater than one. However, (E,E)-2,4-Decadienal significantly declined in content across all three seasons. In the category labeled as MR, volatile compounds exhibited a downward trend in the first and third plantings. Overall, the S treatment reduced the content of Maillard reaction products and degradation products of sulfur-containing amino acids but concurrently increased the content of lipid oxidation products.

Figure 2c showcases the comparison of volatile compounds between PS and CK. In both varieties, over 50% of detected compounds saw an increment in content. In the category FO, only Dodecanal demonstrated a notable upregulation. Conversely, 2-Methyl-naphthalene showed a significant decline across all three growing seasons. In the category RS, a number of volatile compounds in the third growing season manifested a notable increase in this treatment, with pyrazine compounds such as Ethenyl-pyrazine, 2,5-Dimethyl-pyrazine, and 2,5-Dimethyl-3-(3-methylbutyl)pyrazine having VIP values greater than one. Within the category SP, two compounds including Dimethyl trisulfide displayed a notable reduction, reaching significant levels. Overall, compared to CK, the PS treatment could enhance the overall levels of lipid degradation products, while showing a general decline in both Maillard reaction products and sulfur amino acid products, although some pyrazine compounds exhibited more than a two-fold increase.

To further elucidate the variations in volatile compounds between different treatments, the PS treatment was compared with P, as shown in Figure 2d. Generally, only a few volatile compounds exhibited differences exceeding two-fold, yet over 50% of the volatile compounds were upregulated overall. In the category FO, a segment of aldehyde compounds demonstrated a significant decrease in content compared to the P treatment, including compounds like 3,5-Octadien-2-ol, Naphthalene, etc. In the category MR, the PS treatment led to higher pyrazine compound levels compared to P, with compounds like Methyl-pyrazine, 2-Ethyl-6-methyl-pyrazine, etc. showing a significant rise across multiple growing seasons. In the category SP, three compounds including Dimethyl trisulfide demonstrated higher levels than in the P treatment. Consequently, compared to the P treatment, the combined usage of both fertilizers could amplify the levels of Maillard reaction products as well as sulfur amino acid degradation products. Similarly, as depicted in Figure 2e, compared to S, the joint application also elevated the majority of Maillard reaction products and sulfur amino acid degradation products.

In general, the addition of phosphorus and sulfur fertilizers influenced the formation of volatile compounds. In studies on wheat cultivated under different conditions, it was found that the application of sulfur fertilizer enhanced the content of lipid oxidation products such as Nonanal and Maillard reaction products like 2-Ethyl-6-methyl pyrazine, consistent with the results of our experiment [42]. Additionally, the experiment indicated that the individual or combined application of phosphorus fertilizer elevated the content of lipid oxidation products such as E-2-hexenal and (E,E)-2,4-Decadienal. A similar phenomenon was also observed when nitrogen fertilizer was applied [43]. Nikolaou et al. [44] found that nitrogen fertilizer only impacted flavor when applied in conjunction with phosphorus fertilizer. Therefore, we believe that the application of phosphorus fertilizer might affect the carbon and nitrogen balance in plants, similar to the application of nitrogen fertilizer. The resultant variations in carbohydrate, amino acid, and fat content might be responsible for the increase in lipid oxidation products. Furthermore, (E,E)-2,4-Decadienal is considered to have a fried aroma [45]. In our study, the S treatment displayed a higher fatty sensory score, even though there was a declining trend in the content of (E,E)-2,4-Decadienal. This suggests that, under the air-frying method, other compounds such as E-2-Decenal contribute to the fatty flavor profile. Blanda et al. [46] believed that 2-pentylfuran and E-2-Heptenal are associated with off-flavors. However, in this study, while the content of these two compounds increased after fertilizer application, a significant rise in off-flavors was not observed (Figure 1), aligning with Bough’s research on the volatile compounds and sensory characteristics of fifteen varieties [22].

### 3.3. The Influence of Different Fertilization Treatments on the Content of Carbohydrates, Amino Acids, and Fatty Acids

Figure 3 depicts the total reducing sugar content and starch content in tubers under four levels of fertilization. As carbohydrates are highly correlated with quality in potatoes, a consistent trend in content variations was observed across three planting seasons. Compared to CK, the S treatment led to a significant more than one-fold increase in reducing sugar content. The P also resulted in a surge in reducing sugar content. The PS likewise notably heightened the level of reducing sugars in potato tubers. This indicates that the increase in reducing sugar content following combined addition might be associated with the application of sulfur fertilizer. The level of starch content also escalates with the addition of either phosphorus or sulfur fertilizer. After the application of sulfur fertilizer (S), the starch content was 1.3 times that of before the addition, and it could reach up to 2.4 times that of CK following the PS treatment. Hence, the additional application of sulfur fertilizer based on phosphorus fertilizer utilization could foster the accumulation of carbohydrates. Meanwhile, adding phosphorus fertilizer based on sulfur fertilizer could effectively reduce the content of reducing sugars. To further explore the influence of fertilization strategies on carbon and nitrogen, a metabolic network map encompassing amino acid synthesis and major fatty acid biosynthesis revolving around the tricarboxylic acid cycle was constructed (Figure 4).

In comparison to CK, fertilization generally elevated the total content of oleic acid and linoleic acid in potatoes to a certain extent, albeit influenced by varietal effects. Following the addition of sulfur fertilizer (S), the contents of both oleic and linoleic acids significantly increased in the D23 variety. The combined application of sulfur and phosphorus fertilizers exhibited a similar outcome. In the QS9 variety, the P treatment significantly raised the total content of palmitic acid in the tubers. By comparing the PS with the P treatment, it was deduced that the heightened levels of C18 unsaturated fatty acids were the result of the effects induced by sulfur fertilizer addition.

Regarding the impact on free amino acid content, fertilizer addition could effectively diminish the levels of free amino acids in the tubers. The S or P treatment reduced the levels of tryptophan, a member of the phenylalanine family, in the D23 variety. The PS treatment significantly augmented the levels of tryptophan while notably reducing the content of phenylalanine. Proline, being one of the potent antioxidants within plants, exhibited a significant reduction in the D23 variety as fertilizers were applied. However, after the PS treatment, its levels were significantly higher compared to P. Concurrently, other amino acids involved in glutamine metabolism exhibited a similar trend in variation. However, closer to the upstream glycolysis metabolism and amidst amino acids circling the tryptophan metabolism, the content of cysteine showed an increasing tendency with the application of fertilizers. The increment in the contents of the starch, reducing sugars, and fatty acids post-fertilization indicates that the plant’s carbon assimilation metabolic activity is vigorous [35]. Pyruvate, being a crucial substrate for the synthesis of carbohydrates like fatty acids, exhibited a reciprocal trend where downstream amino acid levels decreased and upstream amino acid levels increased, further substantiating this hypothesis. Moreover, compared to P, the PS treatment demonstrated an increment in amino acid content surrounding aspartic acid and cysteine metabolism, proving that upon sufficient phosphorus supply, adding sulfur fertilizer could further enhance the content of sulfur-containing amino acids.

### 3.4. Fertilization Treatments as Key Factors Affecting the Amino Acid and Fatty Acid Contents Influencing the Composition of Volatile Compounds

In an endeavor to delve deeper into the relationship between free amino acids, fatty acid content, and volatile compound content, a correlational analysis was conducted with volatile compounds meeting the screening criteria (FC > 1.5, VIP > 1, and *p*-value < 0.05) along with the fatty acids, free amino acids, and total reducing sugar content. Subsequently, a correlational network graph was constructed utilizing compounds fulfilling the conditions of a correlation coefficient exceeding 0.7 and a *p*-value less than 0.05 (Figure 5).

The findings revealed that compounds such as (E,E)-2,4-Nonadienal and Decanal are closely correlated with fatty acid content. Compared to palmitic acid, oleic acid and linoleic acid had higher correlation coefficients with volatile compounds. This can be attributed to the fact that fatty acids with higher degrees of unsaturation are more prone to oxidative cleavage, facilitating the formation of aldehydes [47]. The addition of fertilizers can enhance the fatty acid content in tubers; hence, in comparison to CK, other fertilization treatments manifested an increasing trend in the total oxidation products of fatty acids. Additionally, Maillard reaction products displayed a high correlation with the content of free amino acids and reducing sugars. Amino acids such as leucine and isoleucine exhibited a higher association with volatile compounds. This is because leucine and isoleucine can synthesize pyrazine compounds under conditions abundant in dicarbonyl compounds [48]. When fertilizers were applied, the overall content level of free amino acids decreased, resulting in a lower pyrazine content in the fertilization treatments compared to the control. Reducing sugars are a vital source of dicarbonyl compounds. In the PS treatment, the decrease in amino acids such as leucine was minimal, and a higher reducing sugar content was maintained, thus sustaining a relatively high level of pyrazine compounds. This manifested as a reduction in the pyrazine flavor compound content compared to CK, although not significantly, and conversely, when compared to P or S, an increase in the pyrazine compound content was also observed.

In general, the relationship between flavor compounds and substrates has been well discussed through the construction of a Maillard reaction simulation model [49,50]. However, potato tubers, being a complex matrix, often present an overall variation in amino acid content. By constructing a correlation network, families of amino acids that most likely affect the levels of volatile compounds can be identified. Leucine, isoleucine, valine, and serine are the larger nodes within the network. Among them, valine and leucine belong to the pyruvate derivative family, while serine and isoleucine are part of the aspartate family. Research has found that phosphorus deficiency simultaneously activates signals for sulfur assimilation, promoting sulfur accumulation and assimilation [41]. Consistent with previous studies [34], the CK treatment showed an accumulation of downstream amino acids in the aspartate family (Figure 4). This led to a significant increase in the content of compounds like Dimethyl disulfide. As a result, in the sensory evaluation for the potato-like characteristic, CK scored higher than the fertilizer-treated groups. Pyruvate acts as a key node compound in the synthesis pathways of carbohydrates and amino acids. After the application of phosphorus fertilizer, the synthesis of the potato starch and fatty acids is enhanced, causing a large amount of pyruvate to be consumed in the form of acetyl-CoA. This weakens the carbon source diverted to the synthesis of amino acids in the pyruvate derivative family [34], resulting in a decrease in the content of valine and leucine. Interestingly, in sensory evaluations, both P and S treatments scored higher in roasted characteristics (Figure 1). Simulated experiments by Kocadağlı et al. [51] suggested that the formation of α-dicarbonyl compounds, such as 2,3-Pentanedione, is key to the formation of tri-substituted pyrazines, and a general increase in amino acid levels does not significantly enhance the content of pyrazine compounds. Hence, these results imply that the increase in reducing sugar content due to fertilizer application is the main reason for the elevated pyrazine compound levels. This also explains why the PS treatment scored the highest in roasted flavor. The influence of fertilization management on volatile compounds reflects the complexity of environmental impacts on flavor quality. Further optimization is needed in the future to determine the most cost-effective fertilizer concentrations.

### 3.5. Fertilizer Application Induces Flavor Compound Alterations and Affects Tuber Flavor Evaluation

Flavor sensory characteristics are constituted by a complex system formed by various volatile compounds. Simple correlation analyses are inadequate in explaining the origins of flavor characteristics within a complex matrix [52]. Hence, a PCA analysis based on sensory evaluation scores and compound contents was constructed, extracting the four principal dimensions with the highest loading values as depicted in Figure 6.

As illustrated in Figure 6b, the CK and the P treatment are separated from the treatments with S and the PS treatment in dimension one. This indicates that the sulfur fertilizer treatment harbors a higher concentration of pyrazine compounds, such as 2-Ethyl-6-methyl-pyrazine and Methyl-pyrazine, which exhibit roasted aromatic compounds [51], consequently conferring a higher score for roasted characteristics. Currently, due to intensive agricultural production and the widespread use of low-sulfur fertilizers, the phenomenon of sulfur deficiency in agricultural soils is becoming increasingly severe on a global scale [53]. Traditional fertilization methods only replenish nitrogen and phosphorus, so the study supports the idea that supplementary sulfur might be a potential method to improve the flavor quality of potatoes. In plants, phosphorus primarily participates in the assimilation of carbohydrates [54]. In peanuts, varieties with a high fatty acid content produce peanut butter with higher overall acceptability [55]. Notably, the CK treatment exhibits a higher concentration of sulfur-containing compounds like dimethyl disulfide, thereby obtaining higher scores for potato flavor characteristics. Furthermore, the results demonstrated in Figure 6d indicate that materials from different fertilizer treatments can be distinguished on dimension four, where the P and the PS treatment have higher contents of Decanal and (E,E)-2,4-Nonadienal. These fatty degradation compounds exhibit fruity and oily aromatic characteristics, thereby imparting a certain fatty flavor trait to the respective treatments. Based on the findings above, future endeavors could explore precisely tuning tuber flavors through the manipulation of key genes responsible for phosphorus and sulfur nutrient uptake.

## 4. Conclusions

Phosphorus and sulfur are essential elements for the growth and development of potatoes, and their application significantly influences the flavor quality of the tubers. Whether applied individually or in combination, phosphorus and sulfur fertilizers can stimulate the accumulation of starch and reducing sugars, fostering the synthesis of fatty acids, while simultaneously reducing the content of free amino acids within the tubers. The variation in the content of primary metabolic products further affects the composition of volatile compounds. The application of phosphorus and sulfur fertilizers facilitates the increase in aldehydic compounds but inhibits the production of sulfur amino acid degradation products and some Maillard reaction compounds. In terms of sensory analysis, compared to CK, fertilization improved tuber acceptance. Both S and P influenced roasted, fatty, and potato-like flavors, as well as overall quality. Notably, when phosphorus fertilizer is applied at 180 kg·ha^−1^ and sulfur fertilizer at 90 kg·ha^−1^ in combination, they can maximize the accumulation of fatty acids, starch, and reducing sugars, exerting the most minimal impact on the free amino acid content in the tubers. This results in the highest concentrations of flavor compounds, achieving an optimal enhancement in overall flavor quality, yielding remarkable outcomes.

## Figures and Tables

**Figure 1 foods-12-03764-f001:**
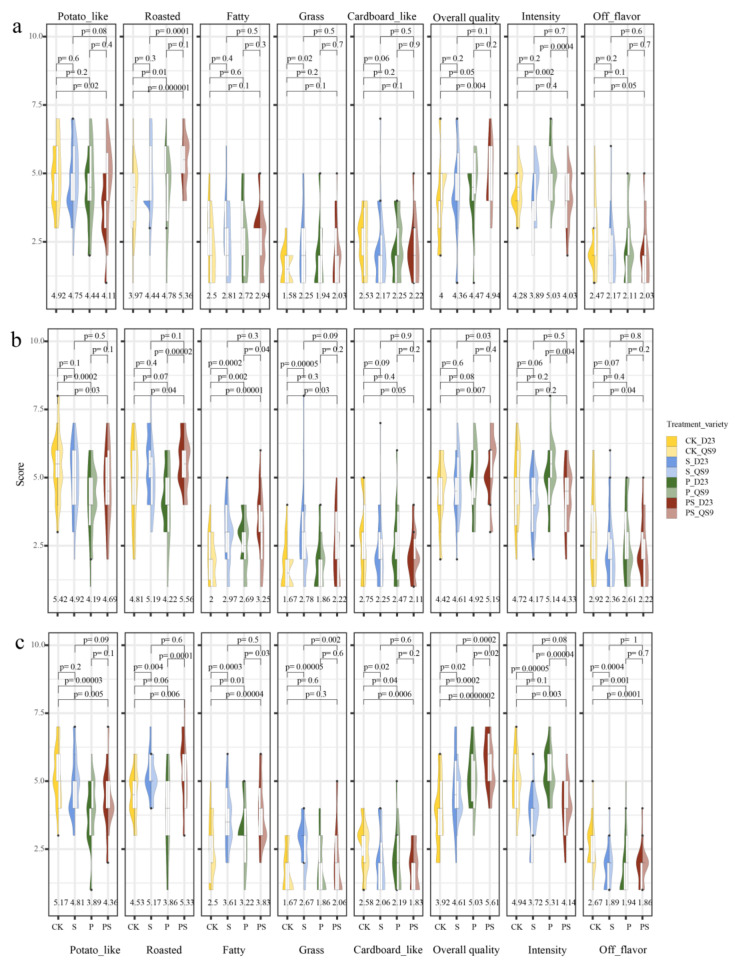
Sensory Evaluation Scores of Tubers under Four Levels of Fertilizer Treatments (D23: Dianshu23, QS9: Qingshu9): (**a**) Distribution of Sensory Evaluation Scores of Tubers Harvested in the First Planting Season; (**b**) Distribution of Sensory Evaluation Scores of Tubers Harvested in the Second Planting Season; (**c**) Distribution of Sensory Evaluation Scores of Tubers Harvested in the Third Planting Season. (“P=” indicates the *p*-value obtained from the significance test between two treatment groups. The left side of each violin plot illustrates the score distribution of Dianshu23, and the right side represents Qingshu9. Numerical values at the bottom of the subplot indicate the average sensory evaluation scores for Dianshu23 and Qingshu9 within the current treatment group).

**Figure 2 foods-12-03764-f002:**
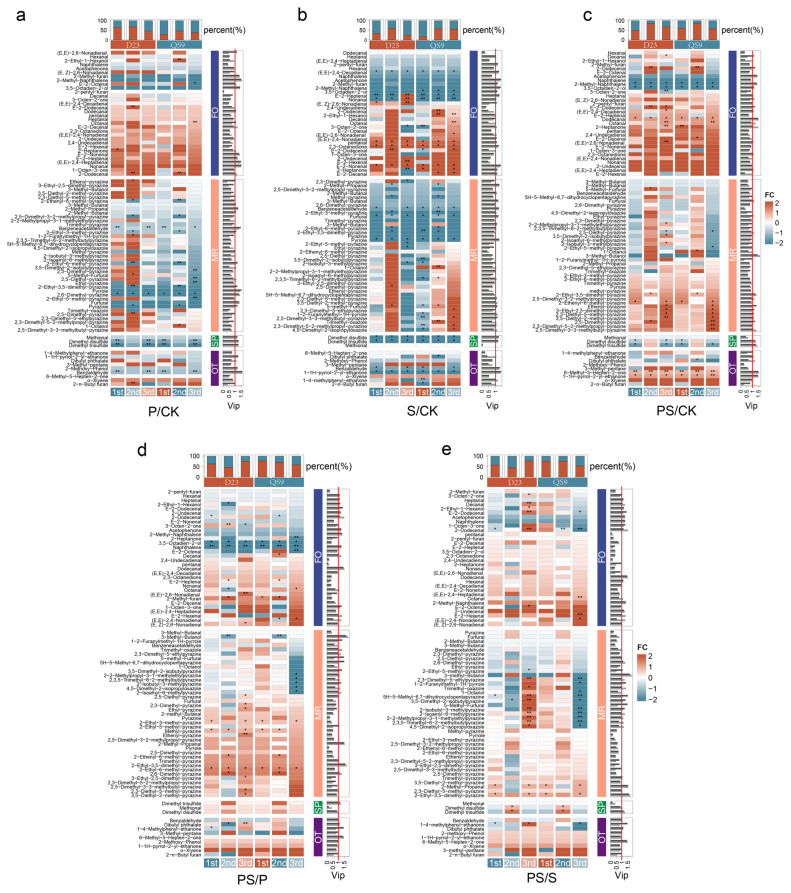
Heatmap Depicting Inter-group Differences in Volatile Compound Concentrations under Four Fertilizer Application Levels (In the heatmap, each row represents a distinct volatile compound; cell color indicates the fold difference, with red denoting an increase and blue denoting a decrease in concentration. Significant differences are indicated by * *p* < 0.05; ** *p* < 0.01. The left three columns represent the Dianshu23 (D23) variety, and the right three columns represent the Qingshu9 (QS9) variety. The legends 1st, 2nd, and 3rd denote the first, second, and third planting seasons, respectively. Side labels represent conjectured compound origins based on the literature: FO indicates potential lipid oxidation products, MR indicates potential Maillard reaction products, SP indicates potential sulfur amino acid degradation products, and OT indicates compounds of unknown origin. The bar graph on the right represents the VIP value acquired from the inter-group OPLS-DA analysis for each compound, with black bars representing the average VIP value across three planting seasons for D23, and gray bars representing that for QS9. The top labels depict the percentage bar charts representing the total proportion of increased (red bars) and decreased (blue bars) concentrations within the current planting season). Subplots: (**a**) P vs. CK; (**b**) S vs. CK; (**c**) PS vs. CK; (**d**) PS vs. S; (**e**) PS vs. P.

**Figure 3 foods-12-03764-f003:**
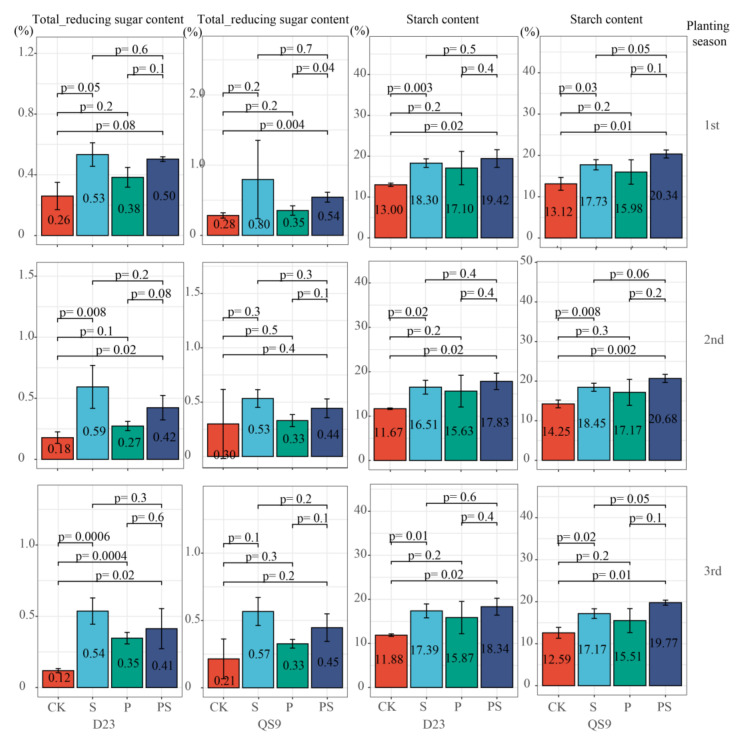
Total Reducing Sugar and Starch Content in Tubers under Four Fertilizer Application Levels (Following “P=” are *p*-values obtained from the significance test between two treatment groups; bar charts represent the average percentage content calculated on a wet basis. D23: Dianshu23, QS9: Qingshu9. 1st: First Planting Season, 2nd: Second Planting Season, 3rd: Third Planting Season).

**Figure 4 foods-12-03764-f004:**
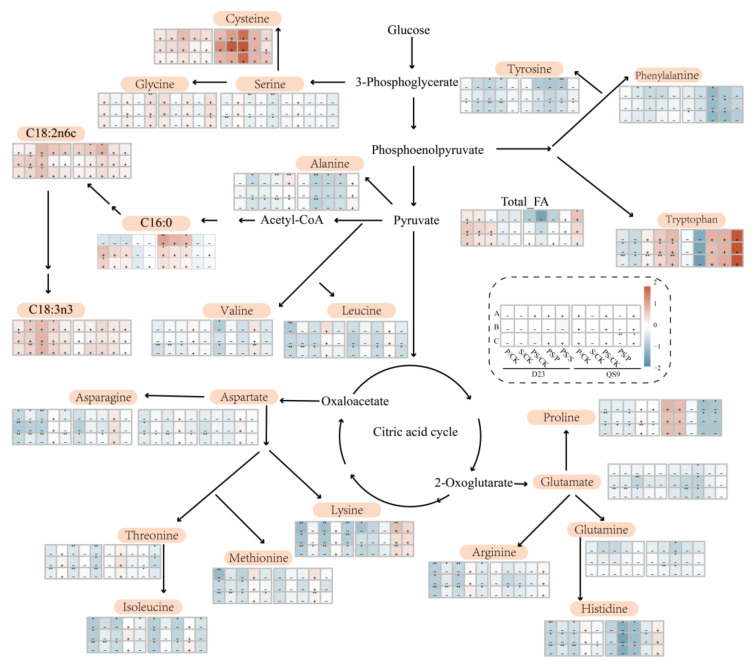
Metabolic Changes of Free Amino Acids and Fatty Acids across Different Fertilizer Treatments (D23: Dianshu23, QS9: Qingshu9; * *p* < 0.05; ** *p* < 0.01; +: relative increase in content level; −: relative decrease in content level).

**Figure 5 foods-12-03764-f005:**
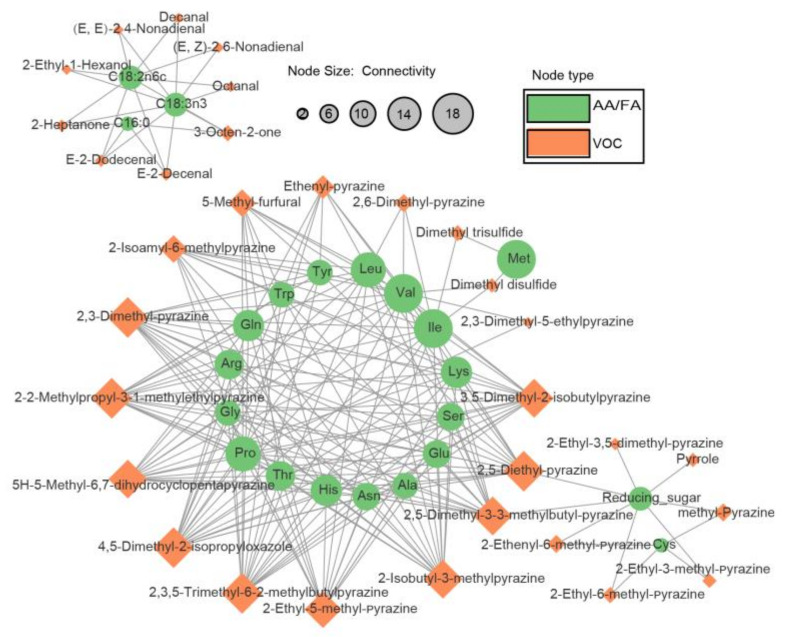
Network Diagram Illustrating the Correlation between Key Differential Volatile Compounds and Fatty Acids, Free Amino Acids across Different Treatments (AA/FA: Free Amino Acids or Fatty Acids, VOC: Volatile Organic Compounds; Connectivity: Number of significantly correlated compounds with the current compound).

**Figure 6 foods-12-03764-f006:**
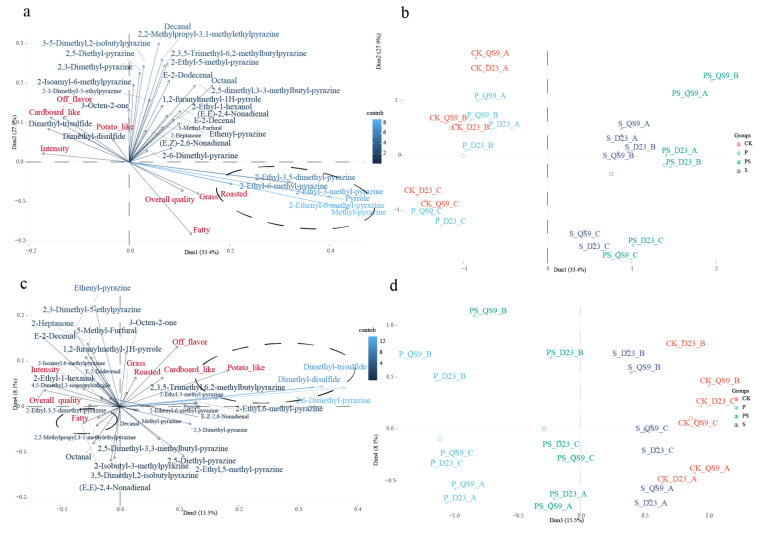
Principal Component Analysis of Volatile Compounds and Sensory Evaluation Scores of Tubers under Four Fertilizer Treatments (Suffixes of the treatments: A: First Planting Season, B: Second Planting Season, C: Third Planting Season; D23: Dianshu23, QS9: Qingshu9): (**a**,**c**) Loading Plots for Different Variables; (**b**,**d**) Principal Component Score Plots for Different Treatments.

**Table 1 foods-12-03764-t001:** Experimental treatment design.

Fertilization Treatment	Abbreviation	P Fertilization Level (kg·ha^−1^)	S Fertilization Level (kg·ha^−1^)
No phosphorus and sulfur application	CK	0	0
Individual phosphorus application	P	180	0
Individual sulfur application	S	0	90
Combined phosphorus and sulfur application	PS	180	90

**Table 2 foods-12-03764-t002:** Chemical properties of experimental soil.

Planting Season	Total P (%)	Available N (mg·Kg^−1^)	Total K (g·Kg^−1^)	Available K (mg·Kg^−1^)	Available S (mg·Kg^−1^)	pH	Organic Matter (%)
1st	2.04	75.98	2.22	17.45	29.1	3.23	3.11
2nd	2.15	75.41	2.15	19.8	32.4	3.15	3.03
3rd	1.93	74.37	2.47	17.8	31.7	3.46	3.17

## Data Availability

Data are contained within the article.

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
