# Peer review of "The Impact of the Individual and Combined Application of Phosphorus and Sulfur Fertilizers on Potato Tuber Flavor"

_foods, 2023, doi:10.3390/foods12203764_

Round 1

Reviewer 1 Report

Please check the attached file for comments and most importantly ensure that ethical approval is provided for conducting this study (sensory section)

Reviewer 2 Report

Dear authors,

Thank you for your great efforts, after a thorough peer-reviewing process, this manuscript had reviewer fascinated by a comprehensive and supporting data, accompanied by a good presentation manner provided by the authors. Only minor revision needs to be made in order to improve manuscript quality.

Great job, congratulation.

Modest proof reading is required to assure this manuscript suits the quality required by the Foods.

Reviewer 3 Report

I send a review of manuscript (ID-Foods–2630684) of the authors: Kaifeng Li , Maoxing Li, Jinhua Zhou, Huachun GuoThe Impact of Individual and Combined Application of Phosphorus and Sulfur Fertilizers on Potato Tuber Flavor”.

I think the manuscript is related to an interesting area of scientific research concerning the indication of the most favorable types and doses of fertilization in the cultivation of potatoes affecting the formation of the content of compounds responsible for the taste and smell of the raw material.

The authors should make a minor revision.

  1. Materials and methods:

General comment:

I would like to ask the Authors to clarify what methods were used to determine the content of reducing sugars and starch? This information was not included in the methodology. (See results in Figure 3 and in discussion)

Page 3, line 105 – Qing Shu 9 or Qingsgu 9 (see. Figure 1)? In these text should be : Dianshu23 (D23) and Qingshu9  (QS9).

Page 3, line 116 – should be K2O

Page 3, lines 124-125; I ask the Authors to explain why the samples of potatoes for sensory evaluation were not given to the evaluators fresh only after storage? For what purpose did the Authors store the tubers at 40C for a week? What about the accumulation of reducing sugars in the tubers?

Page 3, lines 132-133; …eight attributes: fatty, potato-like, ect.…(see under Figure 1).

Page 4, line 145 – (Agilent) please add type and country,

Page 4, line 151 – HP-5ms?

Page 4, line 154 – 50C·min-1, 100C·min-1?

  1. Results and discussion

Page 5, line 197 - please compare the descriptions under Figure 1 as stated inthe methodology on page 3.

Page 9, line 330 - …reducing sugar content and total starch content (This information was not included in the methodology). Please use the term - starch content, without the word total.

Page 10, line 350 – Figure 3…reducing sugar content and starch content…(This information was not included in the methodology).

  1. Conclusions

I ask the Authors to include and indicate the best/beneficial dose and type of fertilization in the conclusions of the study conducted.

References

All literature is cited in the text.
